# Heterostructures formed through abraded van der Waals materials

Darren Nutting[1], Jorlandio F. Felix [2], Evan Tillotson [3,4], Dong-Wook Shin[5], Adolfo De Sanctis [1], Hong Chang[1], Nick Cole[1], Saverio Russo[1], Adam Woodgate[1], Ioannis Leontis[1], Henry A. Fernández [1], Monica F. Craciun[1], Sarah J. Haigh [3,4] & Freddie Withers [1✉]

To fully exploit van der Waals materials and their vertically stacked heterostructures, new mass-scalable production routes which are low cost but preserve the high electronic and optical quality of the single crystals are required. Here, we demonstrate an approach to realise a variety of functional heterostructures based on van der Waals nanocrystal films produced through the mechanical abrasion of bulk powders. We find significant performance enhancements in abraded heterostructures compared to those fabricated through inkjet printing of nanocrystal dispersions. To highlight the simplicity, applicability and scalability of the device fabrication, we demonstrate a multitude of different functional heterostructures such as resistors, capacitors and photovoltaics. We also demonstrate the creation of energy harvesting devices, such as large area catalytically active coatings for the hydrogen evolution reaction and enhanced triboelectric nanogenerator performance in multilayer films. The ease of device production makes this a promising technological route for up-scalable films and heterostructures.

[1] College of Engineering, Mathematics and Physical Sciences, University of Exeter, Exeter EX4 4QF, UK. [2] Instituto de Física, Núcleo de Física Aplicada, Universidade de Brasília—UNB, 70910-900 Brasília, DF, Brazil. [3] National Graphene Institute, University of Manchester, Oxford Road, Manchester M13 9PL, UK. [4] Department of Materials, University of Manchester, Oxford Road, Manchester M13 9PL, UK. [5] Electrical Engineering Division, Department of Engineering, University of Cambridge, 9 JJ Thomson Avenue, Cambridge CB3 0FA, UK. ✉email: f.withers2@exeter.ac.uk

High-quality van der Waals (vdW) heterostructures are produced by stacking together different two-dimensional (2D) materials[1,2]. The properties are highly customisable depending on the component materials and the layer sequence, providing use in a wide variety of applications. Compared with conventional compound semiconductor heterostructure devices, they have the potential to offer many advantages. For instance, they are lightweight, semi-transparent and are compatible with flexible substrates, whilst displaying competitive performance. The highest quality vdW heterostructures out-perform conventional materials, but they are still mainly constructed by mechanical exfoliation of bulk single crystals and built up layer-by-layer by standard mechanical transfer procedures[1–3]. However, this precise yet enormously time-consuming method is not scalable and alternative device manufacturing routes are urgently required to achieve widespread uptake of these materials.

Chemical vapour deposition (CVD)[4] is a promising synthesis approach for vdW heterostructures, where monolayer films are sequentially grown layer-by-layer at high temperatures, with some of the resulting material heterostructures beginning to approach the performance levels of exfoliated crystals. However, the initial investment required and energy cost of CVD growth is high for a given quantity of monolayer material produced. Furthermore, the growth of multilayer systems becomes increasingly complex with the approach confined to a small number of 2D material combinations. Finally, CVD growth requires the use of catalyst substrates and subsequent transfer of the heterostructure films; this often introduces undesirable contamination, tears and cracks which prevent the formation of high-quality vertical heterostructure devices[4].

An alternative low-cost route for mass-scalable production of nanocrystal heterostructures is through printing of liquid phase exfoliated (LPE) dispersions[5–7]. In this scheme, the vdW material dispersions are produced through either ultra-sonication or shear force exfoliation of bulk vdW microcrystals in suitable solvents[8]. This leads to stable dispersions which can then be subsequently printed on a variety of substrates. By mixing the dispersions with specialist binders heterostructures can also be built up layer-by-layer[5]. However, strong disorder in the crystals caused by oxidation, small crystallite size and poor interface quality leads to severe performance degradation compared with devices based on mechanically exfoliated or CVD grown 2D films. In addition, this production method is unlikely to be compatible with the many highly air sensitive vdW materials that are attracting considerable interest recently due to their exotic properties[9,10], limiting the scope of this technology. Moreover, residual solvent in the printed films has been shown to degrade the electrical properties of the devices by further reducing the quality of the interface between neighbouring nanocrystals[11].

This work sets out a route to build up semi-transparent and flexible vdW nanocrystal heterostructures through the simplest possible technique that is through a mechanical abrasion process. Here, we show that high-quality electronic and optoelectronic heterostructures can be readily fabricated within a matter of minutes on the scale of 10s of cm and could easily be scaled up further. The production of rubbed/abraded films have yielded flexible conductive graphite coatings and triboelectric properties in abraded intercalated graphite on steel[12,13]. However, to date, no demonstration of multilayer electronic/optoelectronic devices have been shown. Most surprisingly, using high-resolution scanning transmission electron microscopy (STEM), we observe sharp heterointerfaces formed as a result of the direct abrasion process, which has the potential to facilitate a wide variety of different devices through this approach.

Specifically, in this work we focus on combining several vdW materials including graphite, MoS$_2$, WS$_2$, MoSe$_2$ and hexagonal boron nitride (hBN). In order to highlight the applicability of the abrasive method we show several examples of electronic and optoelectronic heterostructures including thin graphite field effect transistors, vertical transition metal dichalcogenide (TMDC) photodetectors, photovoltaics, hBN capacitors, hydrogen evolution reaction (HER) catalysts and multilayer films for triboelectric nanogenerator (TENG), many of which show significant improvements in device performance compared with those produced by inkjet printing of LPE materials.

## Results

**Device characterisation and fabrication.** The general approach used to produce thin films and devices on SiO$_2$ as well as polymer substrates is shown in Fig. 1a. Essentially, we make use of a viscoelastic polymer, namely polydimethylsiloxane (PDMS), which is cut into 1 cm × 1 cm sections and then pressed into a bulk vdW material powder (graphite, TMDCs, etc). This ensures full adhesion of the micron-sized powder particles to the PDMS surface and allows it to be used as a writing pad. All of the powders investigated adhere equally well to the surface of the PDMS. The PDMS pad is then oscillated back and forth against the substrate with vdW materials embedded between it. We expect that the key parameters which govern the abrasion process on different substrates include the substrate roughness, vdW material hardness[14] and the relative position on the triboelectric series[15] between the vdW material and the substrate (electrostatic charging). Subsequent deposition of material is then due to a friction-facilitated basal cleavage of micro-crystallites within the bulk material powder as it is rubbed against the layers already adhered to the substrate, overall resulting in the deposition of a thin abraded nanocrystalline film. The thickness of the deposited material is controlled by the rubbing time and the force applied to the writing pad. To better quantify the abrasion process we also modified a computer numerical control (CNC) micro engraver system to study the effect of force, feed rate and the sheet resistance vs number of write passes as discussed in "Device fabrication" under "Methods" section and in Supplementary Note 12, Figs. 22 and 23.

To ensure that the vdW material is only written at selective locations, a tape mask can be applied to the substrate before writing (note this is not necessary with the CNC system, unless <5 mm pattern resolution is required). After the design has been written, the tape mask is removed leaving only the unmasked region coated in the vdW material, Fig. 1a (Steps 1–3). This process can then be repeated to build up bespoke heterostructures, Fig. 1a (Step 4). To confirm the structure of our multilayer films we perform STEM energy dispersive X-ray spectroscopy (EDS) elemental mapping of a cross-sectional lamellar, Fig. 1b; ref. [16]. This shows a magnified region of a hBN–Graphite–WS$_2$ heterostructure. The elemental maps reveal the absence of material intermixing, allowing for the formation of heterointerfaces (further STEM images and characterisation can be found in Supplementary Note 1, Figs. 1 and 2). The entire fabrication process was always performed under ambient conditions, although it could easily also be reproduced in a controlled inert gas or high vacuum environment, widening the scope of compatible vdW materials.

An example of a set of connected vertical heterostructures produced through fabrication route 1 is shown in the left of Fig. 1c, with further details of the fabrication steps found in, Supplementary Note 3. A limiting factor on device yield when directly applying a top graphitic electrode was short circuiting, caused by deposition of the top graphitic electrode breaking the barrier material layer underneath. Moreover, we find direct abrasion of graphite onto TMDC's and hBN frequently

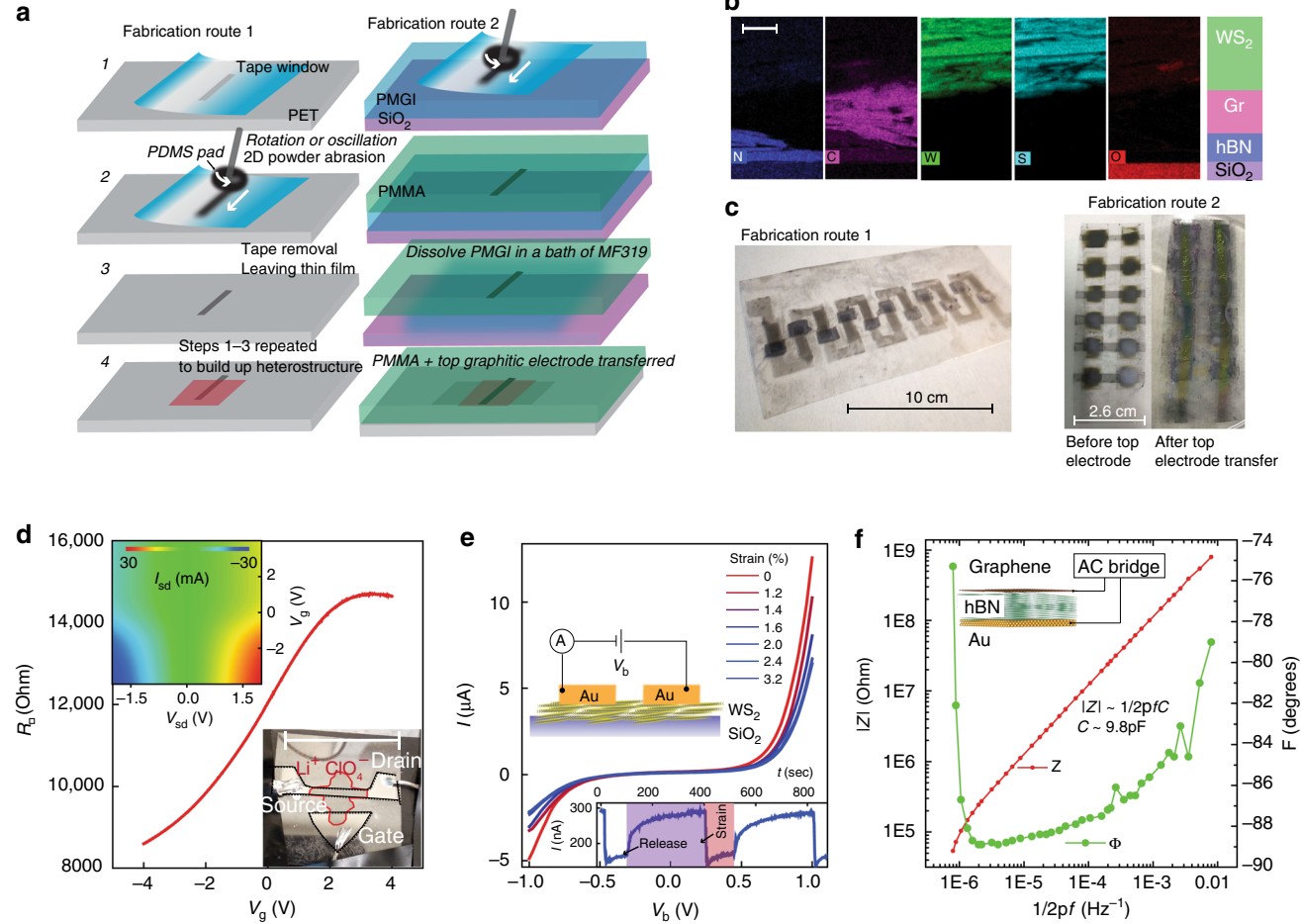

**Fig. 1 Thin films produced through powder abrasion. a** Fabrication routes used to produce heterostructures through mechanical abrasion of vdW powders via a direct write method. **b** STEM-EDS elemental mapping of an abraded vertical heterostructure (scale bar = 400 nm). **c** Left: An example of multilayered vertical junction photodetectors based on a graphite–WS₂–graphite architecture produced via fabrication route 1. Right: The same architecture as the left micrograph but this time following fabrication route two, with the top graphitic electrode transferred from PMGI, which leads to a higher device yield. **d** Gate dependence of the channel sheet resistance for a tape thinned graphitic channel using a LiClO₃ electrolyte (scale bar = 2.5 cm). Top left inset: Contour map of the $I_{sd}$–$V_{sd}$ for different applied gate voltages. Bottom right inset: Optical image of the device. **e** Typical $I$–$V_b$ for a 5 mm × 0.025 mm two terminal planar device based on WS₂ films with a mean film thickness of 1 μm for different levels of applied uniaxial tensile strain. Inset: $V_b$ is held at 0.5 V and the device is subjected to reversible uniaxial tensile strain. **f** Impedance spectroscopy for a hBN dielectric capacitor produced using a 5 μm thick hBN film.

damages the barrier material, likely due to the different materials mechanical properties. While the reverse combination, e.g. TMDCs on graphite are non-damaging. Recent calculations[14] predict that graphite is significantly harder than MoS₂, WS₂ and hBN, which may explain why the former so easily penetrates barrier layers made out of the latter. In order to overcome this, we have also developed a separate fabrication route allowing the successful transfer of the final abraded graphitic top electrode. This is achieved by first abrading the graphite onto a polydimethylglutarimide (PMGI) polymer layer (Fig. 1a, fabrication route 2), before spin coating with polymethyl methacrylate (PMMA). The sacrificial PMGI layer is then subsequently dissolved in a bath of MF319 developer leaving the graphitic film attached to the underside of the PMMA layer, which can then be transferred directly onto the target heterostructure.

After device fabrication we characterised our films through a combination of optical and Raman spectroscopy[17–20], electron transport, atomic force microscopy (AFM) and scanning electron microscopy (SEM) to identify the surface roughness and film

thicknesses (See Supplementary Notes 4, 6 and 7). We find the film roughness, thickness and the Raman spectra to be similar to that seen in liquid phase graphitic films[5,21]. We also provide a study of graphitic film resistance vs transparency which we find to be comparable with sheer force exfoliated films[22]; we expect that the sheet resistance could be further reduced through intercalation methods which enhance the charge carrier density and therefore the conductivity[23]. Our TMDC films on the other hand display similar Raman and optical spectra to bulk or exfoliated crystals[24,25].

Important for use in potential applications are the electronic performance of the films. Interestingly, it is found that the resistance of thin abraded graphitic channels can be controlled by application of a gate voltage. In this case we employ an electrolyte gate, lithium perchlorate (Li⁺:ClO₃⁻)[26], which is drop-cast over the channel region and contacted using a thick abraded graphitic gate electrode, Fig. 1d (inset). The graphite sheet resistance vs gate voltage is presented, with the inset showing a contour map of the current vs source drain bias ($I_{sd}$–$V_{sd}$) for different applied gate voltages. We find the electro-neutrality region to be at large

positive gate voltages indicating strong p-type doping, likely due to ambient water or oxygen doping[27].

It should be noted that not all substrates are compatible with direct abrasion of graphite, we found success with a wide variety of substrates including polyethylene terephthalate (PET), polytetrafluoroethylene (PTFE), PMGI, PDMS, polyethylene napthalate, polyurethane, aluminium (Al), steel and paper but not with $SiO_2$. However, all other vdW materials explored in this work are fully compatible with $SiO_2$ substrates as well. This is likely due to the surface chemistry and roughness of the different substrates, and particularly in how these parameters interact with the hardness of the material being deposited. Unlike previous inkjet printing techniques, no prior treatment of the substrate is required for strong adhesion of the vdW material.

Important for any integrated electronic application is the development of dielectric barriers. Here, we make use of hBN dielectrics produced through mechanical abrasion over evaporated gold electrodes, resulting in film thicknesses of $5 \pm 2\,\mu m$ (estimated from surface profile measurements, Supplementary Fig. 21). Following the deposition of the hBN dielectric, a strip of CVD graphene is transferred onto the hBN film (see "Methods") with two Au electrodes which act as the source and drain contacts for the graphene channel (the schematic of the device is shown in the inset of Fig. 1f). CVD grown graphene is used in order to allow electrostatic gating of the channel region (see Supplementary Note 11). This demonstrates that this technology is also compatible with CVD grown materials and their subsequent transfer. The total area of the capacitor in this instance was estimated to be $2 \times 10^{-6}\,m^2$. The impedance spectrum is presented in Fig. 1f and can be well described by the capacitive contribution, $|Z_T| = (2\pi f C)^{-1}$ at low frequency. The gradient to the linear fit, gives $1/C$ which yields, $C = 9.8$ pF. If we assume a plane plate capacitor model, then the capacitance is related to the dielectric constant, $\varepsilon_r$, by the relation $C = \frac{\varepsilon_r \varepsilon_0 A}{d}$. This allows us to make an estimate of the dielectric constant of the abraded hBN dielectric, which we find to be, $\varepsilon_r = 3 \pm 1$. We note that previous reports have found widely varying values for the dielectric constant of nanocrystal hBN dielectrics with values ranging from 1.5 up to 200[21,28–30], whilst single crystal hBN is known to possess values around ~4[31]. The lower value in our material could be due to air voids in the films lowering the effective capacitance of the whole barrier.

We also performed similar electrical characterisation of vdW heterostructures and films under strain. Figure 1e shows some typical current-bias voltage $(I–V_b)$ curves for a planar Au–$WS_2$–Au channel on a PET substrate, fabricated through shadow mask evaporation with a $25\,\mu m$ channel separation. The different curves are for increasing (red to blue) uniaxial strain generated by bending the 0.5 mm thick PET substrate in a custom-built bending rig (see Supplementary Note 10). We find that the device resistance increases for increased levels of tensile strain, expected as the nanocrystals are being separated. We also find that the resistance changes are highly reversible under both compressive and tensile strain and highly reproducible over $10^3$ cycles (see Supplementary Note 10, Fig. 19). This demonstrates that abraded films could be used for future strain sensor applications.

**Photodetection and photovoltaic devices**. TMDC's are indirect semiconducting materials in the bulk and have already shown great promise for future flexible photovoltaic and photodetection applications[32–35]. Heterostructures based on LPE nanocrystals typically display poor photoresponsivity in the order of 10–1000 $\mu W^{-1}$, restricting their use in practical applications[5,21,36–39].

We explore the use of abraded TMDC materials for photodetection applications in a variety of device architectures, both planar and vertical geometry. Starting with the simplest, we explore a graphitic channel coated with different TMDC's as depicted in the inset of Fig. 2a. This device consists of a tape thinned graphitic channel (required to increase its transparency) with a subsequent layer of TMDC nanocrystals ($MoS_2$, $WS_2$ or $MoSe_2$) abraded on top. Similar double-layered devices have been reported previously and they typically consist of graphene-semiconductor heterostructures[40–42] or graphene hybrid structures such as graphene coated with PbS quantum dots[43]. Essentially, upon illumination photoexcited carriers on the semiconductor transfer to the graphitic layer, resulting in a change of the free charge carrier density leading to a change of electrical conductivity. Our planar photodetectors utilise three different TMDC materials including $MoSe_2$, $MoS_2$ and $WS_2$ abraded onto an ~40% transparent graphitic channel material. Figure 2a shows the temporal response of the photocurrent for the three different TMDC layers with a white light power density of $55\,mWcm^{-2}$ and a bias voltage of $V_b = 2$ V, with the optical excitation aimed through the transparent backside of the PET substrate (enhancing the light incident on the graphite–TMDC interface). The first devices were found to yield responsivities up to $24\,mA\,W^{-1}$ for $WS_2$, constituting more than a $10^2$–$10^3$ improvement compared with other printed LPE photodetectors[36–38,44]. A table comparing our devices and those produced from LPE materials can be found in Supplementary Table 1.

Next we consider a vertical heterostructure geometry consisting of an Au bottom electrode, a TMDC semiconducting barrier and a CVD graphene top electrode. The CVD upper electrode was specifically chosen because of its higher electrical conductivity and optical transparency compared with the abraded graphitic electrodes allowing us to better characterise the optical quality of the abraded TMDC layer.

Figure 2b shows the $I–V_b$ curve in the dark for the device architecture depicted in the inset. A magnified region of the $I–V_b$ curve in the dark and under white light illumination is shown in the bottom right inset, indicating a peak photoresponsivity around $V_b = -1$ V.

The asymmetry in the $I–V_b$ curves here is due to the difference in the work functions of the graphene layer (4.6–4.9 eV)[45] and Au (~5.2 eV)[46] with the conduction band edge of the $WS_2$ closely aligned with the neutrality point of graphene[47]. This means that the conductivity is high at zero bias as electron transport occurs through the conduction band of the $WS_2$[48], while at negative voltages the energy difference between the chemical potential of graphene and the conduction band of $WS_2$ increases, therefore increasing the barrier height and reducing the conductivity. Figure 2c shows the spectral dependence of the photoresponsivity for the same Au–$WS_2$–CVD graphene heterostructure with a peak responsivity found at 2.0 eV, consistent with the peak in absorption associated with the A-exciton in $WS_2$[49]. We find maximal responsivities of $0.15\,AW^{-1}$ at $V_b = -1$ V, again constituting a $10^2$–$10^4$ enhancement compared with printed liquid phase heterostructure photodetectors[5,21,36]. The time response to the incident white light source is also shown in the inset of Fig. 2c, with peak photocurrent values of 100 mA at $V_{sd} = -1$ V. We also explore similar vertical devices based on n- and p-type silicon contacts, which show similar responsivity (see Supplementary Note 5, Fig. 9).

We now move our attention to more complex multilayer vertical heterostructure devices formed through fabrication route 2 (i.e. top abraded graphite films are transferred from PMGI with PMMA support layer) where the entire device comprises abraded films. We focus on graphite–$WS_2$–$MoS_2$–graphite heterostructure diodes with the top abraded graphite electrode mechanically

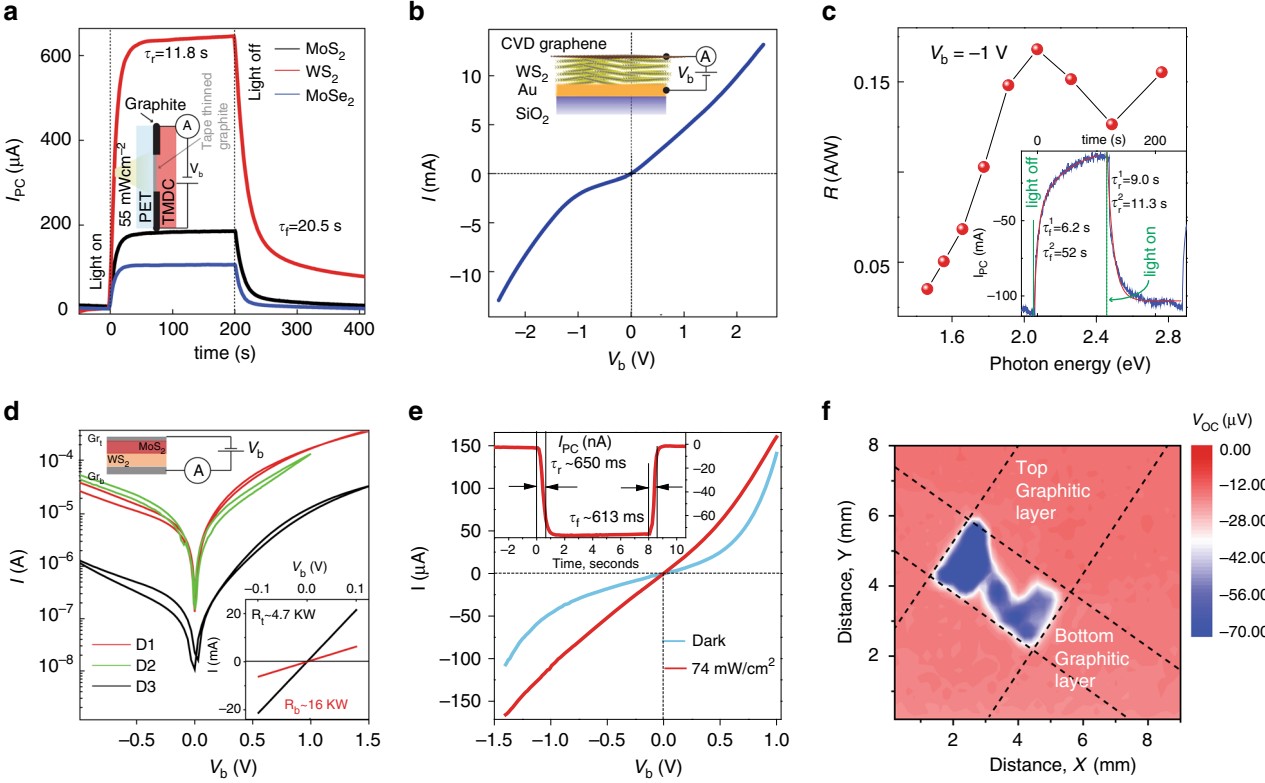

**Fig. 2 Mechanically abraded films for photodetection applications. a** Temporal response for three planar photodetectors abraded onto PET substrates, consisting of graphite–MoS$_2$ (black), graphite–MoSe$_2$ (blue) and graphite–WS$_2$ (red). **b** $I$–$V_b$ for an Au–WS$_2$–CVD graphene top electrode with device area of 1 mm × 1 mm and WS$_2$ film thickness of ~300 nm. **c** Spectral dependence of the photocurrent for the device shown in **b**. Inset: shows the temporal response of the photocurrent with biexponential decay fitted (red curve). **d** $I$–$V_b$ curves for three representative graphite–WS$_2$–MoS$_2$–graphite devices. Inset: $I$–$V_b$ curves for the top and bottom graphitic electrodes. **e** $I$–$V_b$ curves for the device D2 shown in **d** with (red curve) and without (blue curve) white light excitation of 74 mW/cm$^2$. Inset: temporal response of the short circuit photocurrent at $V_b = 0$ V. **f** Photovoltage map of one of our diode structures measured with a focussed laser ($E = 3.05$ eV) with a power output of 0.5 mW and a spot size of diameter 5 μm.

transferred as described above and illustrated in Fig. 1a. Figure 2d shows the $I$–$V_b$ curves of three separate diode devices, all showing very similar behaviour. In total we fabricated and measured 12 junctions, with 10 showing similar electron transport properties. The inset of Fig. 2d shows the $I$–$V_b$ curves of the top and bottom graphitic electrode respectively, showing Ohmic behaviour with typical resistances of a few K Ohm's. Figure 2e shows the $I$–$V_b$ curve for device D2, with and without white light illumination with the bottom right inset of Fig. 2e showing the optical micrograph of the measured device. Such devices offer responsivities between 4–10 mA W$^{-1}$ at $V_b = -1.0$ V, slightly lower than the previous device types likely due to the thicker abraded graphitic top electrode. We also measured the temporal response of the photocurrent as shown in the inset (top right) of Fig. 2e, showing a response times of just ~650 ms owing to the vertical geometry and short channel lengths.

As the white light measurements are obtained when globally illuminating the device, it was important to rule out photocurrent generation from contacts, or elsewhere. To demonstrate this we performed photovoltage mapping measurements with a 405 nm laser beam focussed to a spot size of 5 μm. The photovoltage mapping measurement of a typical device is shown in Fig. 2f, with other devices found in Supplementary Note 5, Fig. 10. We observe a peak open circuit voltage only over the region where all layers overlap, indicating vertical electron transport as the dominant mechanism in these devices. The inhomogeneity in the photovoltage maps arising due to variation of the contact quality of the top graphitic electrodes with the underlying TMDC layer, which

likely explains the order of magnitude reduction in the current for device D3, Fig. 2d.

**Hydrogen evolution reaction (HER).** Mono and few layer TMDC's have been widely studied for their potential use as electrocatalysts for the HER. With recent reports of exceptional HER performance seen in emerging vdW materials[50].

The electrochemical performance of our abraded WS$_2$ films have been characterised in a 0.5 M H$_2$SO$_4$ solution via linear sweep voltammetry (LSV)[51]. To study activity toward HER for catalysts, a three-electrode electrochemical cell was utilised where a PTFE tape was used to define the catalyst area (Fig. 3a). For comparison, a commercial platinum foil with circular area of 0.196 cm$^2$ was also investigated (Fig. 3b, red curve), showing a greater HER activity with a near zero overpotential. The HER polarisation curves of current density are plotted as a function of potential for a representative WS$_2$ film and shown in Fig. 3b (black curve). The onset potential obtained for our WS$_2$ sample was found to be -97 mV (vs RHE). Superior catalyst materials give the highest currents at the smallest overpotential. We find a current density of 10 mAcm$^{-2}$ at an overpotential of 350 mV, comparable with the values observed elsewhere[52–55]. This shows that WS$_2$ films produced through mechanical abrasion are suitable for HER catalyst applications. Figure 3b shows the polarisation curves obtained from just the gold film substrate used to deposit WS$_2$. A noticeable improvement was observed when compared with the gold substrate with the WS$_2$ catalyst,

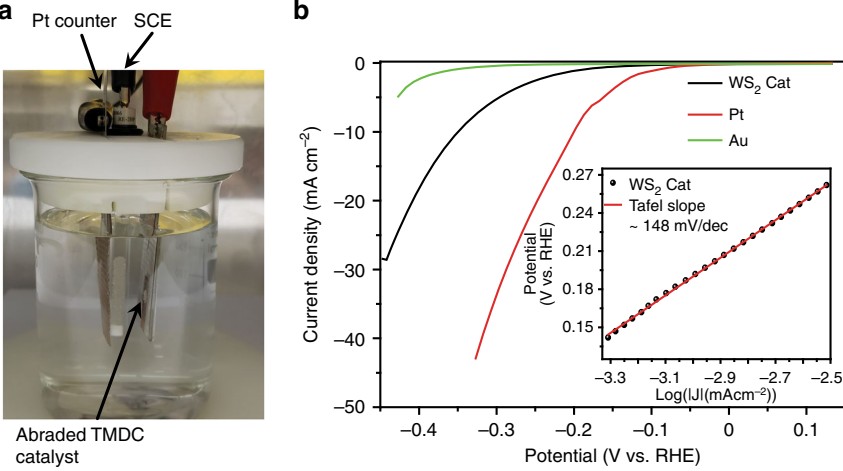

**Fig. 3 WS₂ films as a catalyst for hydrogen evolution. a** Optical micrograph of the electrochemical cell highlighting the different electrodes. **b** Polarisation curves comparing Pt, Au and abraded WS₂ measured in 0.5 M H₂SO₄ with a scan rate of 2 mV/s at room temperature. The inset shows the Tafel plots for our WS₂ sample.

indicating that the catalytic performance is from the TMDC film alone. The overpotential is plotted in the inset of Fig. 3b with the absolute value of the current density within a cathodic potential window and the corresponding Tafel fit shown by the red curve. Thus, the polarisation curve shows exponential behaviour, with the Tafel equation overpotential = $a +$ b log$|j|$ (where $b$ represents the Tafel slope and $j$ is the current density). For our WS₂ films we find a Tafel slope of 148 mV dec$^{-1}$, see inset of Fig. 3b. The reported Tafel slopes for WS₂ films vary significantly for different studies depending strongly on the synthesis route. For example, Bonde et al. reported the HER activity on carbon supported WS₂ nanoparticles with Tafel slopes of 135 mV dec$^{-1}$ [53]. Xiao et al. used an electrochemical route to obtain amorphous tungsten sulphide thin films on nanoporous gold, for which the Tafel slope was 74 mV dec$^{-1}$ [54]. Chen et al. found a similar value (78 mV dec$^{-1}$) for WS₂ prepared at 1000 °C[55]. However, those synthesis routes often involve high temperature processes and/or several steps to obtain the WS₂ catalysts. In contrast, the WS₂ catalysts exfoliated here by mechanical abrasion are rapidly produced through a single low-cost step from cheap and widely available TMDC powders which are already industrially manufactured for lubrication applications[56].

**Triboelectric nanogenerator (TENG).** The triboelectric effect in 2D materials has recently been reported, with previous devices typically being based on thin films produced through liquid phase exfoliation[15,22,57,58]. Here, we demonstrate the use of mechanically abraded thin films and heterostructures as TENG electrodes.

Figure 4a shows a schematic for the operation of a simple TENG charging/discharging cycle using a thin PET substrate and an abraded nanocrystal film or multilayer stack of abraded 2D materials.

Typically, high-quality TENG devices rely on two materials on the opposite end of the triboelectric series[59]. Recently, it has been demonstrated that one strategy for enhancing the power output of a TENG device relies on the use of multilayered structures. In this case, by introducing charge trapping layers such as MoS₂, the magnitude of induced charge per unit area increases leading to enhanced power output[57,58].

To realise a working TENG device, we use an Al hammer wrapped in PTFE tape, with a fluorinated PDMS polymer placed on our abraded vdW electrodes. We compare the performance of abraded graphite to a multilayer graphite/n-type MoS₂ electrode. The operation of the device can be explained as follows: after several contacts between both layers, the PTFE pad is completely released from the PDMS pad, which is in turn attached to the graphite–MoS₂ double-layer, at this point all layers are neutrally charged, Fig. 4a(i) (process 1); Upon approaching the PTFE to the PDMS, electrons are drawn into the graphitic electrode which neutralises the system, resulting in a positive current, Fig. 4a(ii) (process 2); Full contact between these two materials results in charge transfer from one to the other based on the triboelectric series, Fig. 4a (iii) (process 3); Upon releasing, the graphitic electrode is electrostatically induced by the negatively electrified PDMS, and at this moment, free electrons in it move from the graphite electrode to ground, resulting in a negative current, Fig. 4a(iv) (process 4)[15,22,60–62]. To quantify any performance enhancement due to the TMDC trapping layer we compared the response for a simple graphitic TENG electrode to the same graphite layer after coating with a film of MoS₂ (all other experimental parameters were kept the same). Figure 4b shows the generated current through a 1 M Ohm resistor connected in series with the TENG electrodes, for the bare graphitic electrode (black) and the graphite–MoS₂ electrode (red) for several cycles. We found that our first device yields an enhancement of ~50% for the TENG electrode with the MoS₂ trapping layer (the inset shows an optical image of the setup used). After confirming an enhancement due to the TMDC trapping layer we turn our attention to incorporation of the TENG electrode within a practical device. Figure 4c displays the open circuit voltage and short circuit current for three cycles of a secondary, larger device which yields an open circuit voltage in excess of 15 V and short circuit currents of 0.38 μA, giving a peak power output of 5.7 μW, comparable with more complex inkjet pinted TENG electrodes[63,64].

This larger electrode was then used to charge a 10 μF capacitor to 9 V, Fig. 4d. The inset shows the energy stored on the capacitor per cycle (~10 nJ), when connected via a rectifying diode bridge. Given the wide variety of different 2D materials that can be combined we expect the operating efficiency could be significantly improved, thus making abraded 2D materials potential candidates for future flexible energy harvesting TENG electrodes.

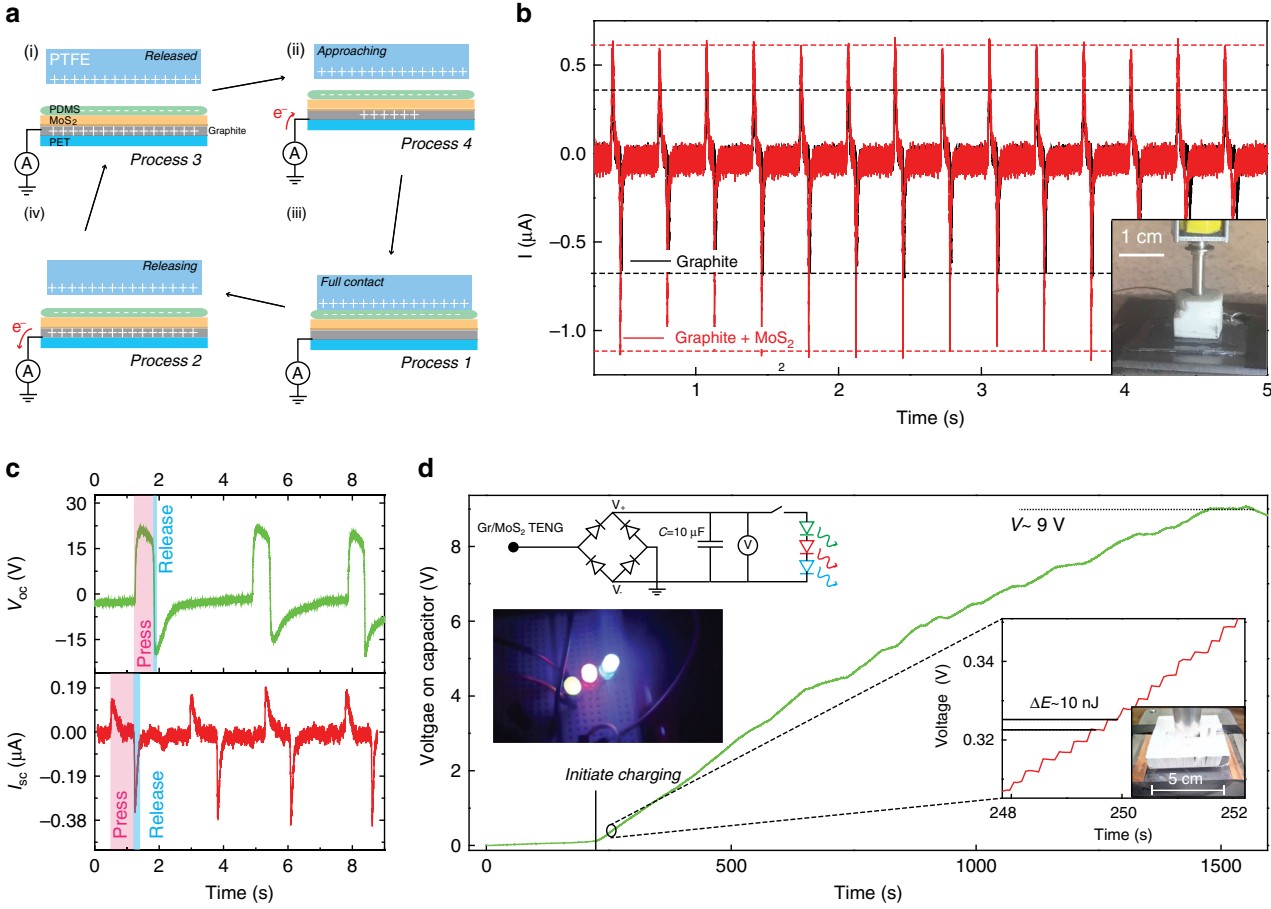

**Fig. 4 TENG films based on abraded van der Waals powders. a** Schematic showing the evolution of charge within the device during a charging/discharging cycle. **b** Current response through a 1 MΩ resistor for an abraded graphite TENG electrode (black) compared with an abraded graphite–$MoS_2$ TENG heterostructure electrode (red). (Inset: PTFE hammer connected to a linear actuator used to generate the voltage pulses). **c** Top: temporal response of the open circuit voltage and bottom: temporal response of the short circuit current, for the graphite–TMDC TENG electrode. **d** Voltage accumulation on a capacitor vs time (hammer frequency ~ 3 Hz). Inset top: rectifying circuit used to charge the capacitor. Inset middle left: three glowing LED's during discharge of the capacitor. Inset right: zoomed in region of the charging curve highlighting the energy stored on the capacitor per cycle.

## Discussion

In this work, we demonstrate the production of multilayer heterointerfaces through the mechanical abrasion of micron-sized vdW crystals on a variety of substrates. We argue that the abrasion works in two stages. Firstly, the deposition of seed layers likely occurs via an electrostatic attraction based on the material and substrates relative position on the triboelectric series (electrostatic charging), the substrate roughness and vdW material hardness. After the seed layer is deposited, the build up of thicker films is then due to friction-facilitated basal cleavage of microcrystals as the powder is rubbed against layers already adhered to the substrate. However, further work is required to understand how each parameter modifies the efficiency of the abrasion process.

Surprisingly, we find that certain combinations of materials can be abraded directly on top of one another resulting in large area heterointerfaces which we confirm through STEM and electron transport studies. We did notice however, that direct abrasion of graphite directly onto softer materials results in damage (confirmed through excessive leakage current in vertical devices or low device yield). This indicates that multilayering should follow a sequence based on the material hardness with the preceding layer being harder than the next to prevent material intermixing and smeared heterointerfaces or device short circuit.

We find that our optoelectronic devices demonstrate significantly enhanced performance compared with LPE materials.

The underlying reason for this is due to larger average crystalite sizes with reduced disorder compared with LPE films[65], we confirm this through analysing the particle size distributions as shown in Supplementary Note 2. This is supported by comparing in and out of plane resistivities of our TMDC films with bulk, exfoliated and LPE crystals. We quantitatively find that our TMDC films display similar resistivity to exfoliated[48,66], CVD[67] or bulk[68] materials, while LPE films display resistivities several orders of magnitude higher than our devices[5,37,38], see Supplementary Table 3.

In conclusion we show that a wide variety of functional heterostructure devices can be built up from 2D nanocrystals through a simple mechanical abrasion method, allowing for rapid up-scaling of heterostructure devices. We demonstrate its practical use in several simple device applications including gate tunable semi-transparent graphitic coatings, hBN capacitors and photodetectors. We have extended the technology and demonstrated the successful creation of various more complex vertical heterostructure devices including multilayer photovoltaics and have shown that abraded $WS_2$ coatings can be used directly as electrocatalysts for HER, as well as demonstrating enhanced TENG electrodes realised through multilayered heterostructures. The ease with which the films can be applied, wide choice of materials, simplicity of up-scalability, low cost and superior performance compared with liquid phase processing makes this

technology significantly attractive for a large variety of future applications.

## Methods

**Materials**. MoS$_2$ (234842-100G), MoSe$_2$ (778087-5G) and graphite (282863-25G) powders were purchased from Sigma-Aldrich. WS$_2$ powder was acquired from Manchester Nanomaterials and the hBN powder was purchased from Momentive (AC6111). CVD graphene on copper foil was purchased from Graphene Supermarket. We used specialised tape (Nitto Denko Corporation) ELP-150E-CM for thinning the abraded films and used both commercial PDMS pads PF-30-X4 (retention level 4) as well as PDMS pads created in-house (SYLGARD 184). The in-house PDMS pads were created by using a 10:1 ratio of silicon elastomer base to curing agent, respectively. These are then mixed thoroughly and left for ~1 h until any trapped air degasses from the mixture before baking at 100 °C for 1 h, or until the PDMS solidifies completely. This baking step is optional and serves to increase the curing speed, as otherwise the mixture will take ~48 h to cure at room temperature. The entire process is completed under ambient conditions, resulting in a pad of elastic modulus ~1.8 MPa[69].

**Device fabrication**. Devices based on mechanical abrasion are fabricated as described in the main text. The thickness of the abraded films can be controlled by the abrasion time and the force applied to the pad used to write the materials on the substrate. To quantify the force, material feed rate and effect of multiple writing passes, we modified a CNC writer to mount the PDMS pad, see Supplementary Note 12. Adjustment of the film thickness via back-peeling with specialist tapes is also possible in order to tailor the transparency/resistance of a particular film. For devices including CVD graphene the fabrication was carried out as follows: PMMA was spin coated onto CVD graphene on copper, a tape window was then attached and the copper etched away in a 0.1 M aqueous solution of ammonium persulfate (APS), which nominally took ~6 h, the CVD graphene was then transferred through two beakers of deionised (DI) water (>8 M Ohm cm) to remove excess APS. The graphene/PMMA membrane was then transferred onto the target device completing the heterostructure. The device along with CVD graphene/PMMA was baked for 1 h at 150 °C to improve the mechanical contact of the CVD graphene with the abraded nanocrystal films. Photodetector devices with amorphous carbon top electrodes were fabricated following methods found in ref. [70]. Photodetector devices with n- and p-type silicon substrates were produced by etching a 1 × 1 cm square of thermally grown SiO$_2$ with a sodium bifluoride etch solution as described elsewhere[71]. The freshly exposed underlying Si was then directly abraded with TMDC powder until no pin-holes were observable under a ×50 microscope objective. A large sheet of CVD monolayer graphene top electrode was then transferred along with PMMA membrane followed by baking at 100 °C for 1 h to improve the contact quality.

**Materials characterisation**. Raman spectroscopy was carried out using 532 nm excitation at 1 mW laser power which is focused onto a 1 µm spot. AFM was performed using a Bruker Innova system operating in the tapping mode to ensure minimal damage to the sample's surface. The tips used were Nanosensors PPP-NCHR, which have a radius of curvature smaller than 10 nm and operate in a nominal frequency of 330 kHz. AFM microscopy images were then analysed using the open source application, Gwyddion[72]. Film thicknesses were measured using an Alpha-Step D-100 Stylus Profiler using minimum force of 0.03 mg. SEM images were obtained using a dual-beam xT Nova Nanolab 600 focussed ion beam (FIB) SEM system. Cross-sectional lamellae were prepared using a Thermo Fisher Helios 660 Dual-Beam FIB SEM. Prior to loading, the samples were coated with a 20 nm layer of high-quality carbon and a 10 nm layer of Au/Pd, providing a uniform conductive coating. The samples were then milled with Ga$^+$ ions of decreasing acceleration voltages and currents (from 30 to 2 keV and from 1 nA to 15 pA, respectively) until electron transparency had been reached. Additional over/under-tilts were required (depending on current) for parallel milling of lamellae without tapering.

**Scanning transmission electron microscopy**. The STEM image data was acquired on an FEI Titan G2 80–200S/TEM operating at 200 kV acceleration voltage. This microscope is equipped with a Schottky field emission gun and spherical aberration probe corrector. STEM data were acquired with a probe current of 380 pA, a semi-convergence angle of 21 mrad and an annular dark field detector inner angle of 64 mrad. EDS STEM elemental mapping was acquired with a 4-EDS detector ChemiSTEM system, a collection solid angle of 0.7 srad, a dwell time of 50 µs and a total acquisition time of 384 s.

**Optical measurements**. Optical transmission spectra were recorded using an Andor Shamrock 500i spectrograph with 300 lines/mm grating resolution and iDus 420 CCD. A fibre coupled halogen white light source was used to excite the photo-active samples which generates 1.4 W at the fibre tip. The white light is collimated to give uniform excitation of 70–100 mW cm$^{-2}$. The white light source was blocked for the time response using a mechanical shutter with a response time of 10 ms. The spectral dependence of the photocurrent was carried out using 10 nm

band pass filters to filter the halogen white light source with the power at each wavelength measured using a Thorlabs photodiode S120C.

**Electrical measurements**. Electron transport measurements were carried out using a KE2400 source-metre for both source and gate electrodes. An Agilent 34410A multimeter was used to record the voltage drop over a variable resistor in order to determine the drain current and photo response for different load resistances. Capacitance spectroscopy was performed using a Rhode and Schwarz, Hameg HM8118 LCR Bridge.

Electrochemical data were obtained using an Ivium-stat potentiostat/galvanostat. LSV experiments were carried out in 0.5 M H$_2$SO$_4$ with a scan rate of 2 mV s$^{-1}$. For determination of activity of HER, a three-electrode electrochemical cell was used, i.e., saturated calomel electrode (SCE) (reference), platinum foil electrode (counter) and WS$_2$/Au (working). The work electrode area used was 0.147 cm$^2$. The reference electrode was stored in KCl solution and rinsed with deionised water before use. For the measurements, high-purity N$_2$ gas was bubbled into the solution for at least 60 min before the electrochemical measurements. The potentials reported here are with respect to reversible hydrogen electrode ($E$ (RHE) = $E$ (SCE) + 0.273 V[54]).

## Data availability

The data that support the findings of this study are available from the corresponding author upon reasonable request.

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

## Acknowledgements

F.W. acknowledges previous support from the Royal Academy of Engineering, the Royal Society grant RG170424 and EPSRC sub-grant EP/S017682/1 Capital Award emphasising support for early Career researchers. J.F.F. acknowledges the Brazilian agencies CNPq (grant number: 430470/2018-5), FAPDF (grant number: 00193-00002066/2018-15 and 193.001.757/2017), for financial support and the research scholarship. D.N. Acknowledges support from the Leverhulme trust. SJH and E.T. acknowledge financial support from the European Union H2020 programme (grant EvoluTEM, 715502) and EPSRC grant EP/P009050/1.

## Author contributions

D.N. developed the transfer process for the graphitic top electrodes, designed (along with N.C.) the custom bending rig, carried out device fabrication, measurement, analysis as well as performing atomic force, scanning electron microscopy, Raman characterisation, developed the CNC production method and contributed to writing the manuscript. J.F.F. carried out the electrochemical measurements, analysis and contributed to the writing of the manuscript. I.L. mixed up the lithium perchlorate electrolyte. A.D.S. provided high-resolution spectral photocurrent measurements and contributed to writing the manuscript. S.R. provided electrical measurement resources. M.F.C suggested to use the films for TENG electrodes and contributed to writing of the manuscript. D.-W.S. developed the TENG measurement setup, interpreted the TENG data and contributed to writing the manuscript. H.A.F. carried out the initial optical transmission spectroscopy. H.C., N.C. and A.W. provided technical support. E.T. and S.J.H. provided STEM characterisation of the multilayer films and contributed to writing the manuscript. F.W. initiated and supervised the project, contributed to sample fabrication, electron transport, optical measurements, analysis and writing the manuscript.

## Competing interests

The authors declare no competing interests.
