## [Peer Review File · Nature Communications]

Reviewers' comments:

Reviewer #1 (Remarks to the Author):

In this manuscript, the authors stated that to fully exploit van der Waals materials and their vertically stacked heterostructures, a new mass-scalable production route, through the mechanical abrasion of bulk powders, was introduced. They successfully achieved significant performance improvements with the abraded heterostructures compared to ink-jet printing of nanocrystal dispersions prepared layers. Different functional heterostructures such as resistors, capacitors, photovoltaics, the creation of energy devices, and enhanced triboelectric nanogenerator have been demonstrated. The results are exciting, but some fundamental problems should be stated clearer to make the work repeatable and convincing before the acceptance for publication.

As many devices have been shown in the manuscript, I do not doubt that there is indeed some van der Waals material film on different substrates and functions very well if the experimental results are correct. However, as the title of this manuscript shows, the novelty of this work is the fabrication of heterostructures through abraded van der Waals materials. The mechanical abrasion is the key protocol to achieve all those high performances of various devices listed in the manuscript. Therefore, it is very important to have a good characterization and explanation of this abrasion process.

1. As abrasion is a type of friction process, therefore, the whole procedure involves the substrate, the van der Waals material, the upper object used to carry out the abrasion, and the working conditions like the environment, load, sliding speed. All those parameters should be given in the MS or the supplementary information. It is too vague only state that "The PDMS pad is oscillated back and forth against the substrate with various 2D".
2. Since abrasion is so important to give a high performance of the various devices, the interface between the van der Waals materials and various substrates should be at a very good contact condition. How did this happen? The abrasion mechanism, friction process and mechanism should be scientifically characterized. Such as under different load, oscillating speed (amplitude and frequency?), how did this van der Waals material film attached well to the substrate in a controlled manner? Because different substrates have been used in different devices, the interaction at the interface is always different. What is the size of the PDMS pad, and what kind of PDMS materials is used? What is the elastic modulus or viscoelasticity of the PDMS?
3. PDMS is coated in a bulk 2D material powder and used as a writing pad. How is it coated, what is the state of those 2D material powder stay on the PDMS substrate? Film thickness, conformation etc. ?
4. As state in the manuscript, the thickness of the abraded films can be controlled by the abrasion time and the force applied to the pad used to write the materials on the substrate, adjustment of the film thickness via back-peeling with specialist tapes is also possible in order to tailor the transparency/resistance of a particular film. The mechanism of those abrasion related fabrications should be reasonably revealed as it's the major contribution to the area of the nanofabrication of heterostructures with van der Waals materials.

Reviewer #2 (Remarks to the Author):

Freddie Withers and coworkers in their work "Heterostructures formed through abraded van der Waals materials" reported a simple and scalable approach to realise a variety of functional heterostructures based on van der Waals nanocrystal films produced through the mechanical abrasion of bulk powders, which appears to be interesting. The following are some remarks which need to be carefully considered:

Comment 1. The authors claimed that the performance improvements can be obtained in abraded heterostructures compared to those fabricated through ink-jet printing of nanocrystal dispersions. However, the test condition should be considered, otherwise, it is meaningless. Thus, it is better to contrast the key performance parameter (e.g. photocurrent, photoresponsivity, response time, EQE etc for the photodetector) of heterostructure device with that of reported in literatures in a table which including the test condition (e.g. bias, light intensity etc for the photodetector).

Comment 2. It was hard to find that the thickness of hBN in Figure S10 was $5 \pm 1\mu\text{m}$. Is it possible to give out a cross-sectional line profiles from the AFM top-views to display the thickness? or a SEM images? In addition, the measuring scale was blurry for the AFM maps in Figure S10.

Comment 3. Please double check their manuscript carefully to avoid some typical mistakes and mistypes, just as example all authors' names were given out in some references while others not.

Comment 4. There are some grammatical and spelling mistakes, such as "whilst" and "at the same time" in line 6 on page 2 was the same meaning, "achiee" in line 11 on page 2, "works" in line 10 on page 3, and so on, please revise them and check the whole text.

Reviewer #3 (Remarks to the Author):

Review of:

Heterostructures formed through abraded van der Waals materials

- General Remarks

The authors present a deposition method for creating networks composed of layered crystal particulates. These networks are created by rubbing bulk crystals onto a substrate whereby the friction presumably shears off the smaller particles. The materials can be layered upon one another to create vertically stacked heterostructures with surprisingly clean interfaces. The authors create several different device types from such heterostructures to demonstrate the versatility of the technique. The TENG in particular is a very good demo. This is an interesting piece of work but the following needs to be addressed before publication here can be considered.

- Primary Issues

The main issue I have here is that no explanation of how the technique works is given anywhere and "abraded" isn't defined anywhere in the text. Can the authors explain what the abrasion process is? Is the crystal shearing apart, basal fracture, etc.? What is the average size of the shorn/fractured crystallites (average length and thickness)? Are these comparable to exfoliated nanosheets? How or how not? What does the size distribution of the crystallites look like?

There is no discussion of why the networks are so good, it's implied that the reader will infer they are through the device demonstrations. At the very least, the conductivities of the various abraded materials could be given (it appears the dimensions are all known?) as a proxy for network quality. This would then give a ballpark to compare actually compare the networks to other exfoliation methods. And how do the in- and out-of-plane conductivities compare? The data appears to be already available from the in- and out-of-plane photodetectors.

The deposition itself is described as a function of abrasion time and force. How is force quantified here? Is it just hand pressure? As these networks are binder-free, one can easily imagine that the networks could be wiped off the substrate by finger pressure, is that right? How much/little

pressure is required to not affect the lower networks when building up a heterostructure? This seems critical.

Am I right in saying that the thickness is controlled by making the networks thick and then using tape to peel material from the top? How is the quantity of peeled material controlled? Is the tape lightly lain on the network? Is it pressed down? How would one go about making a 100 nm network? What's the average thickness for a single abraded line?

What's the problem with the graphite? The authors state that "We find direct abrasion of graphite onto TMDC's and hBN frequently damages the barrier material, likely due to the different materials mechanical properties. While the reverse combination, e.g. TMDs on graphite are non-damaging." The graphite also cannot be abraded on SiO₂ while all other materials can. I know this is probably a question that requires a study to answer but could this be illuminated with a discussion of bulk crystal properties (interlayer binding energies, etc)?

How is the bulk crystal applied to the PDMS writing pad? How are the "in-house" PDMS pads created? Do all crystals adhere similarly or uniformly?

The structure of the planar photodetector in Figure 2 is not clear at all. What am I looking at in the optical image inset? Is the graphite layer split into a source and drain contact in the lower schematic? Unclear. In fact, the figures are heavily overloaded in general, in particular Figures 1 and 2. I'd recommend moving some data to the SI. The inset images are frequently too small to make out anything.

In Figure 2D, there is a ~ 1.5 order of magnitude difference between D3 and D1 & D2. Is this sort of device-to-device variability common across these devices? What's the typical yield? Is the variation related to doing things by hand?

The authors state "The first devices were found to yield responsivities up to 24 mA/W for WS₂, constituting a $\sim 10^3$ - 10^4 improvement compared fully printed 2D-ink photodetectors.³⁰" Hardly a fair comparison given the reference is a fully inkjet printed device and the structure presented here is completely different. Are there not more appropriate references with lithographically deposited contacts?

The authors state "Moreover, residual solvent in the printed films has been shown to degrade the electrical properties of the devices by further reducing the quality of the interface between neighbouring nanocrystals¹¹". Does Ref 11 contain data showing the effect of residual solvent? I can't see it in the main or supplementary figures. In fact, I'm unaware of such data in the literature so I'd be very interested in a reference showing such.

It's probably not worth referring to these networks as 2D films since all the properties are bulk crystal properties and not mono- or few-layer properties. Alsop⁶, for example, refers to "few-layer thick abraded graphitic channels". What does few-layer mean in this context? Is this implying few-layer graphene or a few abraded layers? How does one distinguish the number of "abradings" for the same material. Same for "bilayer" used throughout the text. These adjectives already have specific meanings in the context of these materials.

Figure 1D, the dependent variable is resistance by units but the symbol for resistivity is used. "Resistivity" is used in the caption and "resistance" is used in the main text. Please clarify.

Dear Referee's,

Thank you for taking the time to read our manuscript. The comments/questions have been extremely helpful, and we've tried our utmost to address those concerns in the new version.

Please find a detailed response to the concerns below. Changes within the manuscript have been highlighted in red font.

Yours Sincerely,

F. Withers and D. Nutting

PS. We have managed to setup a small lab at home and can carry out most device fabrication and electrical measurements for this project if more information is required. However, we are unable to carry out optical spectroscopy measurements at present as the university has fully closed and we are unsure when they plan to reopen.

Reviewers' comments:

Reviewer #1 (Remarks to the Author):

In this manuscript, the authors stated that to fully exploit van der Waals materials and their vertically stacked heterostructures, a new mass-scalable production route, through the mechanical abrasion of bulk powders, was introduced. They successfully achieved significant performance improvements with the abraded heterostructures compared to ink-jet printing of nanocrystal dispersions prepared layers. Different functional heterostructures such as resistors, capacitors, photovoltaics, the creation of energy devices, and enhanced triboelectric nanogenerator have been demonstrated. The results are exciting, but some fundamental problems should be stated clearer to make the work repeatable and convincing before the acceptance for publication.

We thank the referee for finding our results exciting. We've tried to address the concerns raised below.

As many devices have been shown in the manuscript, I do not doubt that there is indeed some van der Waals material film on different substrates and functions very well if the experimental results are correct. However, as the title of this manuscript shows, the novelty of this work is the fabrication of heterostructures through abraded van der Waals materials. The mechanical abrasion is the key protocol to achieve all those high performances of various devices listed in the manuscript. Therefore, it is very important to have a good characterization and explanation of this abrasion process.

We've repeated some measurements to confirm photoresponsivity for the Au-WS₂-CVD-Graphene device using a 650 nm band pass filter (Figure S6C). We find that the photoresponsivity is almost the same as the white light measurements shown in Figure 2C confirming the reproducibility of the measurement. The planar graphite-MoS₂ photodetector (Figure 2A) was also measured using a second measurement system and measured by colleagues from a separate research group in Exeter, Figure S11. We obtain a similar responsivity for the MoS₂ device measured in Figure 2. This highlights the reproducibility of the results.

1. *"As abrasion is a type of friction process, therefore, the whole procedure involves the substrate, the van der Waals material, the upper object used to carry out the abrasion, and the working conditions like the environment, load, sliding speed. All those parameters should be given in the MS or the supplementary information. It is too vague only state that "The PDMS pad is oscillated back and forth against the substrate with various 2D".*

We thank the referee for encouraging us to carry out a more systematic and quantifiable approach to the fabrication process. To quantify this large parameter space would likely take an entire study and work is ongoing at present. However, we have made some progress by modifying a computer numerical control (CNC) micro engraver system. This has enabled us to study the effect of force, feed rate and sheet resistance vs the number of write passes under ambient conditions. Initial results are now available within the Supplementary Information section 12.

2. *"Since abrasion is so important to give a high performance of the various devices, the interface between the van der Waals materials and various substrates should be at a very good contact condition. How did this happen? The abrasion mechanism, friction process and mechanism should be scientifically characterized. Such as under different load, oscillating speed (amplitude and frequency?), how did this van der Waals material film attached well to the substrate in a controlled manner? Because different substrates have been used in different devices, the interaction at the interface is always different..."*

We currently postulate that the abrasion process works in two stages. Firstly, the deposition of the initial layers likely occurs via an electrostatic attraction based on the material and substrates relative position on the triboelectric series, substrate roughness and vdW material hardness which are all key parameters affecting the abrasion process. The subsequent deposition of material is then due to friction-facilitated basal cleavage of micro-crystallites within the powder as it is rubbed against the layers already adhered to the substrate. This is now commented on in page 5, of the manuscript. A study into the adhesion of an abraded graphite film is now available within the Supplementary Information, section 13. A study of the abrasion methods effectiveness under different load and feed rate is also available within the Supplementary Information section 12.

".. What is the size of the PDMS pad, and what kind of PDMS materials is used? What is the elastic modulus or viscoelasticity of the PDMS?"

The PDMS pads typically used to abrade van der Waals films by hand were typically cut into 1cm x 1cm squares. This has now been added to page 4, line 7. The process used to create PDMS in-house typically results in a pad of elastic modulus ~ 1.8 MPa, similar to values reported in literature for SYLGARD 184. This information is now available within the Methods section of the manuscript.

3." PDMS is coated in a bulk 2D material powder and used as a writing pad. How is it coated, what is the state of those 2D material powder stay on the PDMS substrate? Film thickness, conformation etc. ?"

Simple light pressing of the PDMS pad into the bulk 2D powder is enough to ensure full adhesion to it and allows for the PDMS to be used as a writing pad. This is now mentioned on page 4, line 8. The state of the material on the pad is in the form of the starting microcrystal before initiating writing. We are not presently able to quantify in detail the film on the PDMS after writing. This will be looked at in detail in a future study where we will study the effects of changing the vast number of different parameters to optimise the processing.

4. "As state in the manuscript, the thickness of the abraded films can be controlled by the abrasion time and the force applied to the pad used to write the materials on the substrate, adjustment of the film thickness via back-peeling with specialist tapes is also possible in order to tailor the transparency/resistance of a particular film. The mechanism of those abrasion related fabrications should be reasonably revealed as it's the major contribution to the area of the nanofabrication of heterostructures with van der Waals materials."

We have added a comment on this in the methods section. The product code of the tape which we used for back peeling is mentioned in the materials section. Back peeling to thin the film was achieved by lightly pressing the tape onto the graphite film and peeling back. A study of sheet resistance vs transparency for a selection of abraded graphite films is now available within the Supplementary Information, section 8, Figure S15.

Reviewer #2 (Remarks to the Author):

Freddie Withers and coworkers in their work "Heterostructures formed through abraded van der Waals materials" reported an simple and scalable approach to realise a variety of functional heterostructures based on van der Waals nanocrystal films produced through the mechanical abrasion of bulk powders, which appears to be interesting. The following are some remarks which need to be carefully considered:

We thank the referee for finding our results interesting. We've tried to address the concerns below.

Comment 1. "The authors claimed that the performance improvements can be obtained in abraded heterostructures compared to those fabricated through ink-jet printing of nanocrystal dispersions. However, the test condition should be considered, otherwise, it is meaningless. Thus, it is better to contrast the key performance parameter (e.g. photocurrent, photoresponsivity, response time, EQE etc for the photodetector) of heterostructure device with that of reported in literatures in a table which including the test condition (e.g. bias, light intensity etc for the photodetector)."

We thank the referee for pointing this out. We've surveyed the literature and made a table in the supplementary information showing a like-for-like comparison of the photoresponsivity of our abraded devices with equivalent liquid-phase exfoliated devices, section 5, Table T1. In general, our devices offer a significant performance enhancement compared to liquid phase devices (10^2 - 10^4) depending on which report. In some cases it was rather problematic to make an exact like-to-like comparison due to the large variability in parameters between device.

However, work is still required to improve the performance further, which could potentially be achieved through optimisation of post fabrication annealing.

Comment 2. "It was hard to find that the thickness of hBN in Figure S10 was $5 \pm 1\mu\text{m}$. Is it possible to give out a cross-sectional line profiles from the AFM top-views to display the thickness? or a SEM images?.."

We thank the referee for pointing out this omission. We have now provided the cross-sectional line profile of this film within the Supplementary Information, section 11, Figure S21a.

"..In addition, the measuring scale was blurry for the AFM maps in Figure S10."

This has now been fixed to improve its legibility.

Comment 3. "Please double check their manuscript carefully to avoid some typical mistakes and mistypes, just as example all authors' names were given out in some references while others not."

The reference listing has now been altered to include all authors' name.

Comment 4. "There are some grammatical and spelling mistakes, such as "whilst" and "at the same time" in line 6 on page 2 was the same meaning, "achieve" in line 11 on page 2, "works" in line 10 on page 3, and so on, please revise them and check the whole text."

The manuscript has been checked again in search of remaining typos.

Reviewer #3 (Remarks to the Author):

Review of:

Heterostructures formed through abraded van der Waals materials

- *General Remarks*

The authors present a deposition method for creating networks composed of layered crystal particulates. These networks are created by rubbing bulk crystals onto a substrate whereby the friction presumably shears off the smaller particles. The materials can be layered upon one another to create vertically stacked heterostructures with surprisingly clean interfaces. The authors create several different device types from such heterostructures to demonstrate the versatility of the technique. The TENG in particular is a very good demo. This is an interesting piece of work but the following needs to be addressed before publication here can be considered.

We thank the referee for finding our results interesting. We've tried to address the concerns below.

- *Primary Issues*

"The main issue I have here is that no explanation of how the technique works is given anywhere and "abraded" isn't defined anywhere in the text. Can the authors explain what the abrasion process is? Is the crystal shearing apart, basal fracture, etc.?."

We are working on this question at present (We have the CNC system and some basic electrical measurement resources in the garage), this will likely take us a few months to complete the differential experiments to help us build the model. We have now included the likely mechanism of the abrasion process in the text on page 5, line 3. However, we need to make an experimental study to find the dominant variable which initiates the process. These variables are substrate roughness, 2D material hardness and the relative position on triboelectric series between the substrate and 2D material. Once the seed layer has been deposited basal fracture and vdW bonding of nanocrystals allows us to build up films/networks to thicknesses in excess of 1 μm .

"..What is the average size of the shorn/fractured crystallites (average length and thickness)? Are these comparable to exfoliated nanosheets? How or how not? What does the size distribution of the crystallites look like?"

Evan and Sarah have carried out detailed crystallite size analysis of a selection of abraded films from cross sectional TEM data. This is now available within the Supplementary Information, section 2, Figure S3, along with a comparison to typical crystallites

produced through liquid phase exfoliation. The text compares to size distributions of the crystals found in liquid phase films. We find that crystallites are slightly thicker than those found in liquid phase dispersions which might be the underlying reason for our improved performance compared to liquid phase devices (properties are more like bulk).

"There is no discussion of why the networks are so good, it's implied that the reader will infer they are through the device demonstrations. At the very least, the conductivities of the various abraded materials could be given (it appears the dimensions are all known?) as a proxy for network quality. This would then give a ballpark to compare actually compare the networks to other exfoliation methods. And how do the in- and out-of-plane conductivities compare? The data appears to be already available from the in- and out-of-plane photodetectors."

We thank the referee for pointing this out. We have added a Table (T3) in the Supplementary Information, section 8. This compares the in and out of plane resistivities of the TMD films to materials produced through a variety of production methods including: mechanical exfoliation, CVD growth and liquid phase and bulk. We find that the resistivity of our TMD films are significantly lower than that found in liquid phase devices.

Our graphite films show comparable sheet resistance and transparency to films produced through shear force exfoliation (S15).

I'm concerned about the validity of these comparisons as the resistivity will depend strongly on carrier density. Unless we are sure the density is the same in each of these reports then this comparison is not valid. We do plan on carrying out Hall measurements on the films as soon as the University reopens. The ballpark figure for our in and out of plane conductivity is closer to bulk or exfoliated few layer crystals than liquid phase exfoliated films.

"The deposition itself is described as a function of abrasion time and force. How is force quantified here? Is it just hand pressure? As these networks are binder-free, one can easily imagine that the networks could be wiped off the substrate by finger pressure, is that right? How much/little pressure is required to not affect the lower networks when building up a heterostructure? This seems critical."

The initial devices detailed in the manuscript were indeed created by hand, and so "force" as mentioned here does refer to hand pressure. From our experience the devices were quite durable and could resist being rubbed between thumb and index finger (as hard as possible) without significant detriment to its performance, although this is hard to quantify. In an attempt to address the question of adherence of the abraded films to the substrate we performed measurements of the sheet resistance before and after submerging in water for 10 minutes (See Supplementary Information, section 13). We observe a factor of 1.76 increase of the sheet resistance indicating that some of the

material is removed, however the vast majority of the film remains adhered to the substrate. This also seemed likely through its apparent lack of change in transparency, but we could not quantify this due to lack of available equipment. However, we did quantify the force, feed rate and abrasion time and its effect on sheet resistance by using a modified computer numerical control (CNC) micro engraver system as is now detailed in the Supplementary Information, section 12.

"Am I right in saying that the thickness is controlled by making the networks thick and then using tape to peel material from the top? How is the quantity of peeled material controlled? Is the tape lightly lain on the network? Is it pressed down? How would one go about making a 100 nm network? What's the average thickness for a single abraded line?"

Typically, we continued to abrade until the sheet resistance got down to ~1-2 KOhm/sq, then the transparency can be tuned by lightly pressing the adhesive tape on the graphite by hand and peeling off some material. For our planar graphite/TMDC photodetectors however, it was important that the graphitic electrodes had regions of increased transparency to facilitate light reaching the graphite-TMDC interface. For this the specialist tape was pressed hard into the film and a thin strip removed. Information regarding the effect of back peeling on the sheet resistance and transparency is available within the Supplementary Information, section 8.

"What's the problem with the graphite? The authors state that "We find direct abrasion of graphite onto TMDC's and hBN frequently damages the barrier material, likely due to the different materials mechanical properties. While the reverse combination, e.g. TMDs on graphite are non-damaging." The graphite also cannot be abraded on SiO₂ while all other materials can. I know this is probably a question that requires a study to answer but could this be illuminated with a discussion of bulk crystal properties (interlayer binding energies, etc)?"

We think that this is due to the varying hardness of the different abraded vdW materials, with graphite being significantly harder than MoS₂, WS₂ and hBN, potentially explaining why for former can easily penetrate the latter but not vice-versa. This is now stated on page 6 of the manuscript (along with a new reference supporting this). This, in conjunction with the differing surface chemistries of the substrates used. This is likely why we found no success abrading graphite on SiO₂ as the SiO₂ is not as rough as the PET. This discussion is now included on page 7 of the manuscript.

"How is the bulk crystal applied to the PDMS writing pad? How are the "in-house" PDMS pads created? Do all crystals adhere similarly or uniformly?"

Simple light pressing of the PDMS pad into the bulk 2D powder is sufficient for good adhesion between the two. This is now stated on page 4 of the manuscript. The explanation of the method used to produce the in-house PDMS is now included in the

Methods section of the manuscript, and section 12 of the Supplementary Information. It appears that all bulk powders investigated here adhere equally well to the PDMS.

"The structure of the planar photodetector in Figure 2 is not clear at all. What am I looking at in the optical image inset? Is the graphite layer split into a source and drain contact in the lower schematic? Unclear. In fact, the figures are heavily overloaded in general, in particular Figures 1 and 2. I'd recommend moving some data to the SI. The inset images are frequently too small to make out anything."

Efforts have now been made to improve the legibility of all figures within both the manuscript and Supplementary Information. With that specific figure the thick graphite regions are the source and drain electrodes ($R_s \sim 100 \text{ Ohm/sq}$), with a tape thinned strip (highlighted in red). The whole graphite film is then abraded with a TMDC film and the device is illuminated through the transparent tape thinned graphite region. Light is incident on the MoS₂-Graphite interface. Photoexcited carriers transfer to the graphite channel increasing its conductivity. The I-V_b curve for the WS₂ device is presented in figure S6A with the device image shown more clearly in the inset.

"In Figure 2D, there is a ~ 1.5 order of magnitude difference between D3 and D1 & D2. Is this sort of device-to-device variability common across these devices? What's the typical yield? Is the variation related to doing things by hand?"

One problem with the vertical devices where we transfer the top graphite electrode is the uniformity of the adhesion. We can see this in the photocurrent maps shown in figure S10. The large variation is therefore likely due to poor uniformity of the contact, so the effective junction area is significantly smaller. This along with some variation of the barrier thickness is the likely reason for the discrepancy.

"The authors state "The first devices were found to yield responsivities up to 24 mA/W for WS₂, constituting a ~10³-10⁴ improvement compared fully printed 2D-ink photodetectors.³⁰" Hardly a fair comparison given the reference is a fully inkjet printed device and the structure presented here is completely different. Are there not more appropriate references with lithographically deposited contacts?"

We were attempting to compare to scalable technologies only. As requested, we have now changed the comparison to more appropriate devices available within the literature. We have included a table in the supplementary information, Section 4 T1 which compares the responsivities to literature values where we can directly compare devices under the same conditions. Our devices typically display a 10²-10⁴ enhancement compared to liquid phase exfoliated photodetectors.

"The authors state "Moreover, residual solvent in the printed films has been shown to

degrade the electrical properties of the devices by further reducing the quality of the interface between neighbouring nanocrystals¹¹". Does Ref 11 contain data showing the effect of residual solvent? I can't see it in the main or supplementary figures. In fact, I'm unaware of such data in the literature so I'd be very interested in a reference showing such."

An appropriate reference has now been given which discusses inhomogeneity at the interface as a result of residual solvent.

It's probably not worth referring to these networks as 2D films since all the properties are bulk crystal properties and not mono- or few-layer properties. Also p6, for example, refers to "few-layer thick abraded graphitic channels". What does few-layer mean in this context? Is this implying few-layer graphene or a few abraded layers? How does one distinguish the number of "abradings" for the same material. Same for "bilayer" used throughout the text. These adjectives already have specific meanings in the context of these materials.

We agree with the referee that the use of the term's bilayer and 2D are somewhat misleading given the nomenclature in the field. We have swapped bilayer for double layer and removed the use of 2D replacing with vdW when discussing the materials.

Figure 1D, the dependent variable is resistance by units but the symbol for resistivity is used. "Resistivity" is used in the caption and "resistance" is used in the main text. Please clarify.

We thank the referee for pointing this out. We were somewhat ambitiously using 2D resistivity. We have now used sheet resistance when discussing the graphitic films.

REVIEWERS' COMMENTS:

Reviewer #1 (Remarks to the Author):

The comments have been considered and made corrections.

Reviewer #2 (Remarks to the Author):

The authors have added relevant information and the manuscript can be accepted.

Reviewer #3 (Remarks to the Author):

The authors have dealt with all criticisms adequately

Dear Referee's

Thank you for taking the time to review our work and for finding it suitable for publication in Nature Communications.

Kind Regards,

Darren Nutting and Freddie Withers